# A Multicenter, Open-Label, Phase I/II Study of FN-1501 in Patients with Advanced Solid Tumors

**DOI:** 10.3390/cancers15092553

**Published:** 2023-04-29

**Authors:** Gary Edward Richardson, Raed Al-Rajabi, Dipesh Uprety, Anis Hamid, Stephen K. Williamson, Joaquina Baranda, Hirva Mamdani, Ya-Li Lee, Li Li, Xingli Wang, Xunwei Dong

**Affiliations:** 1Cabrini Health, Malvern, VIC 3144, Australia; 2University of Kansas Cancer Center, Kansas City, KS 64114, USA; 3Barbara Ann Karmanos Cancer Institute, Department of Oncology, Wayne State University School of Medicine, Detroit, MI 48201, USA; 4Fosun Pharma USA, Princeton, NJ 08540, USA

**Keywords:** FN-1501, FLT3 inhibitor, solid tumors, AML, Phase I/II

## Abstract

**Simple Summary:**

Fms-like tyrosine kinase type III (FLT3) inhibitors have been shown to induce significant clinical responses in patients with acute myeloid leukemia (AML) harboring FLT3 mutations. Responses to monotherapy and combination regimens are typically incomplete and transient, prompting the exploration of novel FLT3 inhibitors in AML as well as solid tumors. The aim of this study was to explore the safety and preliminary efficacy of a novel, intravenous FLT3 inhibitor, FN-1501, in patients with advanced solid tumors. The safety profile was consistent with that of approved FLT3 inhibitors. Preliminary efficacy and safety data from this study support further study of FN-1501 as an alternative treatment for patients with solid tumors.

**Abstract:**

Background: FN-1501, a potent inhibitor of receptor FMS-like tyrosine kinase 3 (*FLT3*) and *CDK4/6*, *KIT*, *PDGFR*, *VEGFR2*, *ALK,* and *RET* tyrosine kinase proteins, has demonstrated significant in vivo activity in various solid tumor and leukemia human xenograft models. Anomalies in *FLT3* have an established role as a therapeutic target where the gene has been shown to play a critical role in the growth, differentiation, and survival of various cell types in hematopoietic cancer and have shown promise in various solid tumors. An open-label, Phase I/II study (NCT03690154) was designed to evaluate the safety and PK profile of FN-1501 as monotherapy in patients (pts) with advanced solid tumors and relapsed, refractory (R/R) AML. Methods: Pts received FN-1501 IV three times a week for 2 weeks, followed by 1 week off treatment in continuous 21-day cycles. Dose escalation followed a standard 3 + 3 design. Primary objectives include the determination of the maximum tolerated dose (MTD), safety, and recommended Phase 2 dose (RP2D). Secondary objectives include pharmacokinetics (PK) and preliminary anti-tumor activity. Exploratory objectives include the relationship between pharmacogenetic mutations (e.g., *FLT3*, *TP53*, *KRAS*, *NRAS*, etc.), safety, and efficacy; as well as an evaluation of the pharmacodynamic effects of treatment with FN-1501. Dose expansion at RP2D further explored the safety and efficacy of FN-1501 in this treatment setting. Results: A total of 48 adult pts with advanced solid tumors (N = 47) and AML (N = 1) were enrolled at doses ranging from 2.5 to 226 mg IV three times a week for two weeks in 21-day cycles (2 weeks on and 1 week off treatment). The median age was 65 years (range 30–92); 57% were female and 43% were male. The median number of prior lines of treatment was 5 (range 1–12). Forty patients evaluable for dose-limiting toxicity (DLT) assessment had a median exposure of 9.5 cycles (range 1–18 cycles). Treatment-related adverse events (TRAEs) were reported for 64% of the pts. The most common treatment-emergent adverse events (TEAEs), defined as those occurring in ≥20% of pts, primarily consisted of reversible Grade 1–2 fatigue (34%), nausea (32%), and diarrhea (26%). The most common Grade ≥3 events occurring in ≥5% of pts consisted of diarrhea and hyponatremia. Dose escalation was discontinued due to DLTs of Grade 3 thrombocytopenia (N = 1) and Grade 3 infusion-related reaction (N = 1) occurring in 2 pts. The maximum tolerated dose (MTD) was determined to be 170 mg. Conclusions: FN-1501 demonstrated reasonable safety, tolerability, and preliminary activity against solid tumors in doses up to 170 mg. Dose escalation was terminated based on 2 DLTs occurring at the 226 mg dose level.

## 1. Introduction

Receptor tyrosine kinases (RTK), a group of transmembrane proteins, are regulators of growth, differentiation, metabolism, and other key functions in normal cells [1,2]. Dysregulated RTK signaling is associated with various human pathologies, including tumorigenesis [3,4]. 

FMS-like tyrosine kinase 3 (FLT3) belongs to the type III RTK family and has a well-established role in the normal growth and differentiation of hematopoietic precursor cells [5,6]. FLT3 dimerizes and auto-phosphorylates upon binding of FLT3 ligand (FL), activating the intracellular tyrosine kinase domain that induces conformational changes followed by activation of intercellular signaling cascades, such as RAS/RAF/MAPK, PI3K/AKT/mTOR, and JAK/STAT pathways that control cellular proliferation, growth, and survival [7,8]. The FLT3 gene is the most frequently altered gene in acute myeloid leukemia (AML) with mutations occurring in up to 30% of patients, where higher FLT3 expression has been associated with poor overall survival (OS) [9,10]. Alterations in FLT3 have also been found in colon adenocarcinoma, lung adenocarcinoma, and invasive ductal breast cancer and to a lesser extent other solid tumors, notably melanoma (11.04%), colorectal cancer (7.91%), endometrial cancer (7.68%), non-small cell lung carcinoma (4.18%), sarcoma (3.92%) and other cancer types in the cancer genome atlas (TCGA) database as analyzed on cbioportal [11,12,13]. Unlike AML, the role of FLT3 gene anomalies has yet to be established for solid tumors. 

FLT-3 inhibitor monotherapy has been shown to achieve complete responses (CRs) and prolong survival in relapsed/refractory (R/R) FLT-3 mutant AML and to a lesser extent, FLT3-wild type AML [14,15]. Insights into the role of FLT-3 biology and pathophysiology have ushered in new paradigms for treatments tailored to heterogeneous AML subtypes with resulting improvements in survival rates, particularly for patients harboring FLT mutations [9,16]. While FLT3 inhibitors may contribute to significant responses achieved in the treatment of AML, primary and secondary resistance mechanisms typically limit clinical benefit [17,18].

Tyrosine kinase inhibitors (TKI’s) targeting FLT3 have been approved for the treatment of various solid tumors (e.g., sorafenib for renal cell carcinoma (RCC), hepatocellular carcinoma and thyroid cancer, sunitinib for RCC and gastrointestinal tumors) but their role in the treatment of AML has not been established [19,20]. Nevertheless, sorafenib generated compelling results in a trial where it was tested against a placebo in conjunction with standard chemotherapy in patients with newly diagnosed AML. While the sorafenib arm of the trial was shown to have significantly prolonged event-free survival (EFS), the impact on overall survival was not statistically significant. Even so, EFS benefits were observed regardless of FLT3-internal tandem domain (FLT3-ITD) mutation status, with greater event- and relapse-free survival accruing to the FLT3-ITD positive subgroup [21,22]. These data suggest that FLT3 targeting may contribute to improved outcomes in the setting of multi-kinase inhibition of AML.

FN-1501 is a tyrosine kinase inhibitor that targets several kinases, including FLT3, cyclin-dependent kinase 4/6 (CDK4/6), platelet-derived growth factor receptor (PDGFR), anaplastic lymphoma kinase (ALK), and the RET protein. Data generated from biochemical, in vitro, and in vivo studies suggest that FN-1501 might provide therapeutic benefits to cancer patients via the inhibition of multiple tyrosine kinases. On the strength of mechanistic and preclinical data, a first-in-human (FIH), open-label, Phase 1/2 dose-escalation, and expansion study was designed to assess the therapeutic potential of FN-1501 in patients with advanced solid tumors or R/R AML [23]. The primary objective of this study was to evaluate the safety and tolerability of FN-1501. The secondary objective was to assess the pharmacokinetics (PK) of FN-1501 and its major metabolites and to examine the preliminary anti-tumor and anti-leukemic activity of FN-1501.

## 2. Materials and Methods

### 2.1. Study Design and Participants

This Phase 1/2, FIH, open-label, multi-center, single-arm, dose-escalation study was conducted in pts with histologically or cytologically confirmed advanced solid tumors or R/R AML. The study was conducted from July 2018 to February 2022 across three sites in the US and Australia. 

Pts eligible to participate in the study underwent a screening period of up to 21 days followed by the investigational agent at doses starting at 2.5 mg/day IV on Days 1, 3, 5, 8, 10, and 12 in 21-day continuous treatment cycles. The study followed a standard 3 + 3 dose escalation scheme using a modified Fibonacci sequence until the maximum tolerated dose (MTD) was reached. Doses increased in 33% increments starting at 22.5 mg/day (cohort 1). As FN-1501 was supplied in 10 mg/vials, each 33% dose increment was rounded to the nearest vial size, such as 5, 10, 15, 22.5 mg/day, etc. The study had a total of 13 dose escalation cohorts. Dose escalation continued until ≥2 dose-limiting toxicities (DLTs) were observed in a cohort of three to six pts, at which point the next lower dose level was declared the MTD. DLT assessments were conducted during the first cycle of treatment with FN-1501.

Among 64 pts enrolled in the trial, 16 failed screening and 48 (75%) were treated with FN-1501 at one of the dose levels tested. Eligible pts were ≥18 years of age with (a) relapsed or refractory AML that had exhausted or was ineligible for therapeutic options, or with ≥5% blasts in the bone marrow according to World Health Organization (WHO) 2016 criteria; or (b) solid tumors evaluable per RECIST v1.1 that had exhausted (or were ineligible for) therapeutic options likely to convey clinical benefit. Other inclusion criteria included Eastern Cooperative Oncology Group (ECOG) performance status (PS) of <2 and life expectancy ≥3 months. Pts with a clinically controlled central nervous system (CNS) tumor or metastasis and adequate organ function were eligible to participate. Ongoing use of strong CYP3A inhibitors or inducers was exclusionary unless participants were willing and able to change to use of an equivalent medication without this profile. Other exclusion criteria included current or recent major surgical procedures, anticancer therapy (chemotherapy, immunotherapy), biologic, live virus vaccine, or investigational therapy within 21 days prior to initiation of the study treatment. For patients in the AML cohort, acute promyelocytic leukemia (APML), chronic myelogenous leukemia in blast crisis (BCR-ABL), or active CNS leukemia were exclusionary. 

The first pt. in each dose cohort was evaluated for any unexpected acute toxicity occurring within 48 h of dosing on Cycle 1 Day 1 (C1D1) dosing. In the absence of worrisome toxicity, the remaining pts could be enrolled and dosed in that cohort. Safety evaluations were performed by investigators, the medical monitor, and the Sponsor. A Safety Monitoring Committee determined dose levels to be administered during dose escalation based on data available from the current and previous dose levels. 

### 2.2. Pharmacokinetics

For PK assessment, blood samples were collected pre-dose and 0.083, 0.25, 0.5, 1, 2, 4, 6, 8, 12, 24, and 48 h post-dose on Cycle 1, Day 1, and pre-dose on Cycle 2, Day 1. Plasma concentrations of FN-1501 and its metabolite M3 were analyzed at KCAS in Shawnee, KS, using a validated liquid chromatography/tandem mass spectrometry method. Primary pharmacokinetic (PK) endpoints were derived from FN-1501 and M3 concentration–time profiles and were analyzed using standard noncompartmental methods of analysis (Phoenix version 8). 

### 2.3. Statistical Methods 

Descriptive statistics were used for all safety, PK, and efficacy parameters. Data were summarized using descriptive statistics (number of pts, mean, median, standard deviation, minimum, and maximum) for continuous variables and using frequencies and percentages for discrete variables. Time-to-event variables were summarized using Kaplan–Meier methods and figures for the estimated median time.

## 3. Results

### 3.1. Patient Disposition 

A total of 48 pts were enrolled and treated in the study (Table 1). All pts who received at least one dose of study treatment were included in the Safety Population. The primary reason for treatment discontinuation was disease progression, as determined by RECIST criteria for solid tumors (27 (42%) pts). Another reason for treatment discontinuation was the infeasibility of lengthy infusion times at RP2D in continuous treatment cycles.

### 3.2. Demography and Baseline Characteristics 

Participants enrolled across three sites in the US and Australia had a median age of 65 years (range: 30 to 92 years) and an ECOG performance status of 0 or 1. Ninety-eight percent of the pts had solid tumors and 2% of the pts had AML. In addition to “other” tumor types, the most frequently reported primary tumor site was ovarian (9 (19%) pts). Overall, the median number of prior systemic treatments was 5 (range: 1 to 12). Baseline demographics and participant characteristics are shown in Table 2. 

### 3.3. Safety and Tolerability 

Two (50%) pts in the 226 mg/d cohort reported ≥Grade 3 AEs that qualified as DLTs. These consisted of Grade 4 thrombocytopenia (1 (25%) pt.) and Grade 3 infusion-related reaction (1 (25%) pt.). Both events resolved and were assessed as probably related to the study treatment. The MTD was subsequently determined to be 170 mg/d. A total of 381 treatment-emergent adverse events (TEAEs) were reported, with at least one TEAE reported in 45 pts. TEAEs are summarized in Appendix A. The TEAEs that occurred in ≥2 pts are presented in Table 3. The most frequent TRAEs were gastrointestinal (GI) disorders, occurring in 30 (63%) pts, consisting of nausea in 15 (31%) pts, diarrhea in 12 (25%) pts, vomiting in 10 (21%) pts, constipation in 7 pts (15%) and abdominal distension in 4 pts (8%). The most frequently reported severe (≥Grade 3) TEAEs were hyponatremia (4 (8%) pts), followed by diarrhea and small intestinal obstruction (each in 3 (6%) pts), and fatigue, pneumonia, and back pain (each in 2 (4%) pts). The remaining severe TEAEs were reported in single pts (Appendix A).

### 3.4. Pharmacokinetic Profile of FN-1501

Forty-eight patients were included in the PK analyses. Overall exposure increased with the increase in the FN-1501 dose. From 2.5 to 226 mg, C_max_ for FN-1501 and metabolite M3 increased with dose in a roughly dose-proportional manner (Table 4 and Appendix A). However, AUC increased in a greater than dose-proportional manner for both FN-1501 and M3 (Figure 1). Exposure for M3 was generally less than 5% of FN-1501. The terminal half-life of FN-1501 and its metabolite M3 ranged from 12.7 to 18.7 h and from 1.43 to 19.8 h, respectively. 

### 3.5. Efficacy Results 

One (25%) patient in the 40 mg/day dose cohort achieved the best response of partial response (PR) by RECIST lasting 4.2 months (efficacy evaluable (EE) population). Fifteen (31%) of patients had stable disease (SD) and eighteen (38%) had progressive disease (PD) as the best response by RECIST (Figure 2). The median progression-free survival was 1.8 months (95% confidence interval (CI): 1.4, 2.7) months. At 3, 6, 9, and 12 months, the progression-free survival was 31.1% (95% CI: 17.6, 45.7), 11.3% (95% CI: 3.7, 23.8), 5.7% (95% CI: 1.0, 16.5), and 2.8% (95% CI: 0.2, 12.5), respectively. The median overall survival was 8.2 months (95% CI: 3.2, not evaluable (NE)) months. The PFS (95% CI) at 3, 6, 9, and 12 months was 71.7% (95% CI: 54.1, 83.5), 59.1% (95% CI: 39.4, 74.2), 44.3% (95% CI: 16.7, 69.0), and 44.3% (95% CI: 16.7, 69.0), respectively (Appendix A). 

## 4. Discussion

This Phase I/II dose escalation study enrolled patients with solid tumors and AML to evaluate the safety and tolerability of FN-1501. While the trial was also designed to enroll pts with R/R AML, the only pt. enrolled and treated in that group was evaluable for safety but not for efficacy. Preliminary data from 48 study participants demonstrated an acceptable safety profile for FN-1501. Fatigue and nausea were the most frequently reported TEAEs, which were manageable and reversible. Two patients in the 226 mg dose group experienced DLTs: one Grade 4 thrombocytopenia and one Grade 3 infusion-related reaction. No DLTs were reported in pts who received doses of 170 mg or lower. The most frequently reported severe (≥Grade 3) TEAEs were non-hematological, including hyponatremia, diarrhea, small intestinal obstruction, fatigue, pneumonia, and back pain, events that were the most common causes for dose reduction and/or dose interruption. In general, the toxicity profile of FN-1501 appeared to be consistent with that of other agents in the class, such as midostaurin, sorafenib, and quizartinib [24,25,26,27].

Overall, FN-1501 appeared to be reasonably safe and well-tolerated. Most AEs were Grade 1 or 2 in severity and primarily consisted of fatigue, nausea, diarrhea, and abdominal pain. Grade 1 or 2 events of fatigue were reported in 21% of pts and nausea in 19% of pts.

Pharmacokinetic data at low doses ranging from 2.5 to 128 mg have demonstrated approximate total body clearance estimated to be 80 to 100 L/h, which is close to or slightly higher than the hepatic blood flood rate in humans. This extreme first-pass effect made FN-1501 unsuitable for oral administration and IV infusion was subsequently chosen as the route of administration. At higher doses, the trend showed dramatically decreased clearance of 50 and 38 L/h at 170 and 226 mg doses, respectively, possibly due to enzyme saturation. In light of these findings, the intermittent dosing schedule (TIW dosing for 2 weeks in 3-week cycles) and moderate half-life, the drug concentration at the end of each dosing interval (either 48 or 72 h) was relatively low at approximately 20 ng/mL at the 48 h post-dose assessment. Nevertheless, with a 30% unbound fraction, the unbound concentration remains close to or slightly higher than the in vitro IC50 obtained for two FLT3-ITD mutation cell lines: MV-4-11 and MOLM-13 (IC50 of 2.6 and 5.2 ng/mL) (data on file, Fosun Pharma). By comparison, the US FDA-approved FLT3 inhibitor, gilteritinib has an IC50 of 0.5 ng/mL for an MV-4-11 cell line. At the approved dose of 120 mg PO QD, gilteritinib has an unbound average concentration at steady state of (Css,u) 15 ng/mL [28,29]. In the MV-4-11 cell line, Css,u/IC50 was 30. For FN-1501, Css,u at 260 mg is 37 and 25 ng/mL for 48 h and 72 h dosing intervals, respectively. This translates to Css,u/IC50= 14 or 10 for the MV-4-11 model. For MV-4-11 and MOLM-13 xenograft mouse models, tumor growth reached stasis at 5 and 20 mg/kg, respectively. Css,u in mice at 5 mg and 20 mg are roughly 5 and 50 ng/mL, respectively. Therefore, exposures achieved at 226 mg, the highest dose assessed in humans, were expected to be within the efficacious range for AML. One caveat is that AML in humans resides in bone marrow, whereas the xenograft AML model more resembles solid tumors. Drug distribution may also affect efficacy. In the dose escalation part of the study, only one AML patient enrolled, limiting any ability to assess the preliminary efficacy of FN-1501 in that disease setting. Only a handful of trials have tested FLT3 inhibitors in solid tumors and none have determined a developmental path in this disease setting. To date, FLT3 targeting is only possible for the treatment of AML.

The best response to FN-1501 primarily consisted of 1 PR lasting 4.2 months (advanced endometrial cancer) and stable disease (31% of pts with advanced ovarian, endometrial, and colorectal cancer, Figure 2) lasting 18–372 days. The remaining response-evaluable pts (67%) were PD.

In summary, our study demonstrated FN-1501 to be safe and well tolerated at doses up through the identified RP2D IV TIW for 2 weeks, followed by 1 week off treatment in continuous 21-day cycles. The study also exposed distinct challenges for clinical development, namely limited single-agent clinical activity and the lack of a biomarker for therapeutic response. Prohibitively long infusion times TIW and limiting pharmacologic properties of FN-1501 mandate the need for oral formulation to achieve and sustain anti-tumor activity. Accordingly, the trial was terminated early in pursuit of more feasible and effective delivery by mouth.

## 5. Conclusions

FN-1501 was tested in patients with advanced solid tumors using a tyrosine kinase inhibitor with significant potency against FLT3, a target that led to the development of agents that have profoundly advanced the standard of care for FLT3-mutated AML (e.g., midostaurin, gilteritinib). Lengthy, tri-weekly infusion times required to deliver sufficiently high doses of FN-1501 limited the ability to determine whether this finding extends to FLT3-mutated solid tumors. Demonstrated safety, tolerability, and preliminary efficacy data generated on the current trial support formulation of FN-1501 that enables convenient delivery in order to further test the therapeutic value of a TKI with significant impact on FLT3 and in biologically appropriate tumor genotype(s).

## Figures and Tables

**Figure 1 cancers-15-02553-f001:**
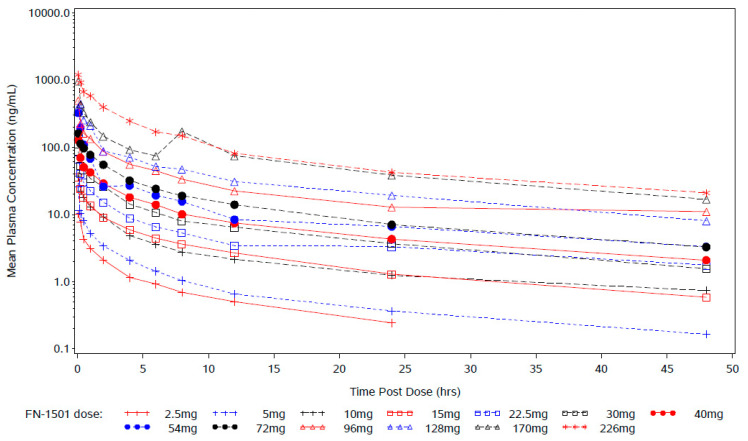
Steady-state plasma concentration profile of FN-1501. Mean plasma levels were measured over 48 h post-dose for all dose levels on Cycle 1 Day 1 and pre-dose on Cycle 2 Day 1.

**Figure 2 cancers-15-02553-f002:**
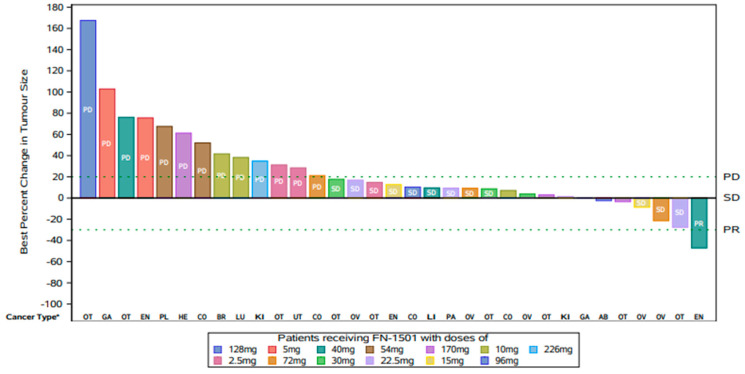
Waterfall plot showing best percentage change from baseline in lesion size and best response for target lesions of all evaluable patients, with tumor type. Each bar represents a single pt. (Abbreviations Efficacy: PD = progressive disease, SD = stable disease, PR = partial response; Cancer Type*: AB = abdomen, BR = breast, CO = colorectal, EN = endometrium, GA = gastric, HE = head/neck, KI = kidney, LI = liver, LU = lung, OV = ovarian, PA = pancreas, PE = peritoneum, PL = pleura, UT = uterus, OT = other).

**Table 1 cancers-15-02553-t001:** Patient disposition and dose levels.

Dose Level	Dose	Number of Patients	Dose-Limiting Toxicity
1	2.5 mg/day	4	0
2	5 mg/day	3	0
3	10 mg/day	5	0
4	15 mg/day	3	0
5	22.5 mg/day	4	0
6	30 mg/day	3	0
7	40 mg/day	4	0
8	54 mg/day	3	0
9	72 mg/day	3	0
10	96 mg/day	5	0
11	128 mg/day	3	0
12	170 mg/day	4	0
13	226 mg/day	4	2 *

* DLT: dose-limiting toxicity (Grade 4 thrombocytopenia (1 pt.) and Grade 3 infusion-related reaction) (1 pt.).

**Table 2 cancers-15-02553-t002:** Patient characteristics.

Characteristic	Number	Percentage
All	48	100
Gender		
Male	21	44
Female	27	56
Ethnicity		
Caucasian	41	85
African American	5	10
Asian	2	4
Median age, years (range)	65 (30–92)	
Primary tumor		
Ovarian	9	19
Colorectal	7	15
Endometrial	3	6
Gastric	2	4
Head/neck	1	2
Uterine	1	2
AML	1	2
Other	10	21
Median number of prior therapies (range)	5 (1–12)	

**Table 3 cancers-15-02553-t003:** Treatment-related adverse events in at least 2 pts.

	FN-1501 Dose (mg/day) IV TIW × 2 Weeks in 21 Day Cycle													
	2.5 (N = 4)	5 (N = 3)	10 (N = 5)	15 (N = 3)	22.5 (N = 4)	30 (N = 3)	40 (N = 4)	54 (N = 3)	72 (N = 3)	96 (N = 5)	128 (N = 3)	170 (N = 4)	226 (N = 4)	Total (N = 48)
Total number of TEAEs	20	23	25	41	26	16	21	12	15	33	43	60	46	381
Total patients with at least 1 TEAE	4	3	4	3	4	3	4	3	2	5	2	4	4	45
Fatigue	1	1	2	0	2	1	1	2	0	0	1	4	0	15
Nausea	1	1	1	2	1	1	0	0	1	2	1	1	1	15
Diarrhea	0	1	0	2	0	1	1	0	1	1	0	2	3	12
Vomiting	1	1	1	0	1	0	1	0	0	2	0	1	2	10
Abdominal pain	1	1	0	1	2	1	0	0	1	0	1	1	0	9
Constipation	0	1	1	1	1	0	0	0	0	1	0	2	0	7
Dizziness	0	0	0	2	1	0	0	0	0	1	1	1	1	7
Infusion-related reaction	0	1	0	1	0	0	0	1	0	1	0	2	1	7
Back pain	1	0	0	1	0	0	0	0	1	0	1	0	2	6
Decreased appetite	1	0	0	0	2	0	0	0	0	0	1	1	1	6
Dyspnea	1	0	1	0	1	0	0	0	0	0	1	2	0	6
Oedema peripheral	1	1	0	0	0	0	1	0	1	0	1	0	0	5
Pyrexia	0	1	0	1	1	0	1	0	0	0	1	0	0	5

Abbreviations: N = number of patients in the dose cohort; TEAE = treatment-emergent adverse event.

**Table 4 cancers-15-02553-t004:** Pharmacokinetics of FN-1501 at each dose level (IV administration).

Dose Level (mg)	AUC_∞_ (h × ng/mL)	AUC_last_ (h × ng/mL)	t_1/2_ (h)	MRT_∞_ (h)	C_max_ (ng/mL)	t_max_ (h)	CL (L/h)	Vss (L)	Vz (L)
2.5	32.4 ± 13.1	29.4 ± 11.9	12.8 ± 3.5	11.7 ± 4.59	12.0 ± 1.42	1.2 ± 0.10	85.1 ± 26.8	913 ± 206	1510 ± 421
5	56.5 ± 18.2	52.1 ± 18.2	18.6 ± 2.34	13.7 ± 3.74	25.5 ± 17.3	1.1 ± 0.02	93.9 ± 25.2	1320 ± 619	2570 ± 932
10	129 ± 41.3	119 ± 32.6	13.7 ± 2.68	12.5 ± 2.84	36.9 ± 17.5	1.2 ± 0.15	84.7 ± 32.4	1110 ± 664	1720 ± 925
15	153 ± 8.45	138 ± 10.8	16.6 ± 3	16.1 ± 2.64	35.6 ± 2.15	1.2 ± 0.14	98.5 ± 5.29	1600 ± 331	2370 ± 536
22.5	239 ± 55.4	251 ± 87.8	18.3 ± 0.49	16.4 ± 1.01	65.8 ± 24.5	1.2 ± 0.19	96.9 ± 22.5	1600 ± 466	2570 ± 663
30	341 ± 47.1	297 ± 55.8	12.4 ± 7.43	13.4 ± 8.49	87.8 ± 37.5	1.1 ± 0.11	89.1 ± 13.3	1130 ± 619	1500 ± 786
40	499 ± 122	437 ± 77.2	18.7 ± 5.68	18.3 ± 6.19	128 ± 41.9	1.1 ± 0.044	83.6 ± 18.6	1450 ± 185	2160 ± 195
54	868 ± 181	773 ± 180	19.6 ± 6.44	16.9 ± 7.34	323 ± 55.6	1.1 ± 0.05	63.9 ± 12.4	1080 ± 516	1820 ± 740
72	823 ± 292	742 ± 275	17.2 ± 1.58	16.8 ± 1.12	161 ± 67.7	1.3 ± 0.24	94.2 ± 28.8	1580 ± 489	2370 ± 833
96	1650 ± 229	1420 ± 236	16.1 ± 6.28	15.4 ± 6.58	494 ± 289	1.1 ± 0.13	59.1 ± 8.20	921 ± 425	1410 ± 682
128	1980 ± 1120	1730 ± 850	17.8 ± 7.00	18.1 ± 5.53	533 ± 344	1.1 ± 0.11	79.9 ± 4.21	1320 ± 473	1790 ± 366
170	3760 ± 1980	3340 ± 1500	15.0 ± 3.64	15.8 ± 7.89	955 ± 330	1.3 ± 0.26	56.5 ± 31.5	757 ± 218	1120 ± 419
226	6070 ± 1670	5620 ± 1390	14.2 ± 2.2	15.2 ± 3.92	1180 ± 232	1.2 ± 0.02	39.4 ± 0.17	569 ± 30.8	787 ± 139

N = number of patients with data. mg milligrams, Tmax time at maximum concentration, h hours; Cmax maximum concentration, ng nanogram, mL milliliter, AUC0-24 area under the curve from time 0 to 24 h, AUC0-48 area under the curve from time 0 to 48 h, AUC0-inf area under the curve from time 0 to infinity, Vz/F apparent volume of distribution, CL/F apparent total clearance, t1/2 terminal elimination half-life.

## Data Availability

Additional data are available in the Appendix A. Datasets are available upon reasonable request from the corresponding author.

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
