# Peer review of "A Multicenter, Open-Label, Phase I/II Study of FN-1501 in Patients with Advanced Solid Tumors"

_cancers, 2023, doi:10.3390/cancers15092553_

Round 1

Reviewer 1 Report

This paper aimed to discuss the safety, tolerability, pharmacokinetics, and anti-tumor activities according to phase I/II clinical treatment of FMS-like tyrosine kinase type III (FLT3) inhibitor, FN1501, on 48 patients with solid tumors. The study is well-designed and thoroughly discussed. However, one major part of this paper emphasized in its abstract and introduction is FLT3 mutation is a potential therapeutic target and highly associated with overall survival, yet, the FLT3 mutations have not been established in this paper. This missing part needs to be established and discussed. 

Author Response

We thank the reviewer for their meaningful comments which have led to an improved and revised manuscript. Please see the attachment.

Reviewer 2 Report

In the manuscript „A multicenter, open-label, Phase I/II study of FN-1501 in pa- 2

tients with advanced solid tumors and acute myeloid leukemia“ authors report on the safety and tolerability of FN-1501, a multiple tyrosine kinase inhibitors. Further, they report of its anti-tumor effect in patients with solid tumors. 

Broad comments: The manuscript is readable and reports on the important matter but there are some suggestions to improve the value of the manuscript

Specific comments: 

1.     There is only one patient with AML in this study so I advise to remove acute myeloid leukemia from the title of the manuscript.

2.     Line 44 – write what DLT is.

3.     Lines 71-76: please rewrite to be more readable, eg “…ductal breast cancer notably breast cancer…” Percentages are in relation to what?  

4.     Line 76 – any recent data on FLT3 gene anomalies in solid tumors?

5.     There is no mention of important guidelines from ELN especially when mentioning management of AML, relapsed/refractory disease etc… (PMID: 35797463)

6.     Lines 93-95: Any recent data of FLT3 targeting in solid tumors?

7.     Table 1 – there is no name of the table. Please rearrange tables to be less confusing- it is rather difficult to follow half the table on one page and second half on another.

8.     Discussion – the authors mention AML cell lines in regard to FLT3 inhibition -  since there is only one AML patient in this study I would advise to focus here more on solid tumor cell lines  

9.     Conclusion brings nothing new to the manuscript

10.  Although I am aware that FLT3 inhibition is largely studied in AML, I would advise authors to focus more on solid tumors throughout the manuscript.

Author Response

We thank the reviewer for their helpful comments. This has led to an improved revised manuscript. Please see the attachment.

Round 2

Reviewer 2 Report

I have no further suggestions.